# Age-Related Properties of Aquaponics-Derived Tilapia Skin (*Oreochromis niloticus*): A Structural and Compositional Study

**DOI:** 10.3390/ijms24031938

**Published:** 2023-01-18

**Authors:** Nunzia Gallo, Alberta Terzi, Teresa Sibillano, Cinzia Giannini, Annalia Masi, Alessandro Sicuro, Federica Stella Blasi, Angelo Corallo, Antonio Pennetta, Giuseppe Egidio De Benedetto, Francesco Montagna, Alfonso Maffezzoli, Alessandro Sannino, Luca Salvatore

**Affiliations:** 1Department of Engineering for Innovation, University of Salento, Via Monteroni, 73100 Lecce, Italy; 2Institute of Crystallography, National Research Council, 70125 Bari, Italy; 3Department of Biological and Environmental Sciences and Technologies, University of Salento, Via Monteroni, 73100 Lecce, Italy; 4Department of Cultural Heritage, University of Salento, Via Monteroni, 73100 Lecce, Italy; 5Typeone Biomaterials Srl, Via Vittorio Veneto, 73036 Muro Leccese, Italy

**Keywords:** Tilapia, skin, aquaponic, collagen extraction

## Abstract

In the last two decades, fisheries and fish industries by-products have started to be recovered for the extraction of type I collagen because of issues related to the extraction of traditional mammalian tissues. In this work, special attention has been paid to by-products from fish bred in aquaponic plants. The valorization of aquaponic fish wastes as sources of biopolymers would make the derived materials eco-friendlier and attractive in terms of profitability and cost effectiveness. Among fish species, Nile Tilapia is the second-most farmed species in the world and its skin is commonly chosen as a collagen extraction source. However, to the best of our knowledge, no studies have been carried out to investigate, in depth, the age-related differences in fish skin with the final aim of selecting the most advantageous fish size for collagen extraction. In this work, the impact of age on the structural and compositional properties of Tilapia skin was evaluated with the aim of selecting the condition that best lends itself to the extraction of type I collagen for biomedical applications, based on the known fact that the properties of the original tissue have a significant impact on those of the final product. Performed analysis showed statistically significant age-related differences. In particular, an increase in skin thickness (+110 µm) and of wavy-like collagen fiber bundle diameter (+3 µm) besides their organization variation was observed with age. Additionally, a preferred collagen molecule orientation along two specific directions was revealed, with a higher fiber orientation degree according to age. Thermal analysis registered a shift of the endothermic peak (+1.7 °C) and an increase in the enthalpy (+3.3 J/g), while mechanical properties were found to be anisotropic, with an age-dependent brittle behavior. Water (+13%) and ash (+0.6%) contents were found to be directly proportional with age, as opposed to protein (−8%) and lipid (−10%) contents. The amino acid composition revealed a decrease in the valine, leucine, isoleucine, and threonine content and an increase in proline and hydroxyproline. Lastly, fatty acids C14:0, C15:0, C16:1, C18:2n6c, C18:3n6, C18:0, C20:3n3, and C23:0 were revealed to be upregulated, while C18:1n9c was downregulated with age.

## 1. Introduction

Type I collagen is the most abundant vertebrate structural protein [1,2,3,4,5]. Thanks to its well-known structural and non-structural properties, it is the most widely used biomaterial for healthcare applications, including use in the medical, cosmetic, nutraceutical, and pharmaceutical sectors [2,6,7,8,9]. In the last two decades, fisheries and fish industries by-products (i.e., skin, bone, scales) have attracted interest as low-cost raw materials for type I collagen extraction [10,11,12,13]. Advantages such as biocompatibility, biodegradability, bioactivity, weak antigenicity, and low transmissible diseases risk have defined fish-derived collagenous materials as safe, alternative, and abundant biomaterials [8,14,15,16,17]. The recovery of discards, that accounts for approximately 70–85% of the total weight of a catch [18,19,20], gives the possibility to valorize the huge quantity of wastes and reduce environmental pollution due to fish processing. Moreover, the request for Halal materials has made fish collagen even more attractive [21].

Among fish species, Nile Tilapia (*Oreochromis niloticus*) is one of the most farmed fish species in the world, accounting for a global production of 6.5 million tons in 2018, with aquaculture production increasing 11% per year [22]. The valorization of aquaponic Tilapia-waste polluting by-products as sources of biopolymers would make the derived materials not only eco-friendlier but also particularly attractive in terms of profitability and cost effectiveness [23]. 

Among by-products, skin was revealed to have a high collagen content and accounts for 28–40% of the tissue dry weight [24,25,26,27]. In accordance with this, Tilapia skin is the most commonly used extraction source. However, to the best of our knowledge, no studies have been carried out to investigate, in depth, the structural and compositional age-related differences in fish skin with the final aim of selecting the most advantageous fish size for collagen extraction, by also improving the scalability of the process. In this study, the effect of age on Tilapia skin was investigated. In particular, skin proximate composition was determined using the AOAC (Association of Official Analytical Chemists) procedure. The amino acid and lipidic profiles were investigated by Gas Chromatography coupled with Mass Spectrometry (GC–MS). Skin ultrastructural analysis was performed by means of Scanning Electron Microscopy (SEM) and by Small- and Wide-angle X-ray scattering (SAXS and WAXS), analyzing the collagen structure from the sub-molecular to the nanoscale. The thermal behavior was determined by Differential Scanning Calorimetry (DSC). The tissue’s three-dimensional organization was analyzed by micro-Computed Tomography (mCT). Lastly, the tissue’s mechanical response was assessed by uniaxial tensile test.

## 2. Results

### 2.1. General Composition

Fish skin general composition was examined by the AOAC procedure to investigate the effect of age (Table 1). Results were in agreement with recent studies on Tilapia skin, where contents of approximately 50–70% water, 20–40% proteins, 2–15% fat, and 0.1–12.0% ashes were registered [28,29,30]. In particular, small Tilapia (ST) and big Tilapia (BT) were found to have a reduced water content compared to farmed Tilapia [28] that was found to be similar to better unspecified Tilapia skin harvested from local markets [28,29,30,31]. Similarly, ST and BT protein and lipid contents were found to be higher than those of farmed Tilapia or those derived from two different local markets [28,29,30] but lower than those found by dos Santos Rodrigues et al. [31]. However, it should not be neglected that literature data comparison is difficult because of the lack of information about fish age and breeding conditions.

Strong differences were found between aquaponic-derived Tilapia skin composition profiles. Indeed, ST and BT proximate composition was found to be strongly influenced by age. In particular, water and ash contents were found to be directly proportional to age, with a significative increase of approximately 13% for the water content (ST: 51.6 ± 3.2%; BT: 64.4 ± 2.4%) and of approximately 0.6% for the ashes content (ST: 0.12 ± 0.03%; BT: 0.70 ± 0.5%) (*p* < 0.01) with age. Ashes increment could be due to the higher calcification of the scales according to the fish age, as reported by Muyonga et al. for young and adult Nile perch [32]. Conversely, the protein and lipid contents were found to decrease according to age (*p* < 0.01). The protein content decreased approximately 8% from 38.9 ± 1.7% in ST down to 30.6 ± 2.1% in BT. The lipid content decreased approximately 10% from 16.5 ± 3.8% in ST to 4.4 ± 0.1% in BT, contrary to Muyonga et al., in which an increase in lipids was registered with age [32].
ijms-24-01938-t001_Table 1Table 1General composition of ST and BT skin in comparison with literature data about Tilapia skin composition. Results are expressed as mean value ± SD.CompositionSTBT[28][29][30][31][33][34][34]Water %51.6 ± 3.264.4 ± 2.472.5 ± 2.164.1 ± 1.071.7 ± 0.357.7 ± 4.172.6 ± 2.576.5 ± 3.071.0 ± 0.6Proteins %38.9 ± 1.730.6 ± 2.122.8 ± 0.421.9 ± 0.121.3 ± 1.241.4 ± 1.221.3 ± 2.565.5 ± 3.360.8 ± 8.1Lipids %16.5 ± 3.84.4 ± 0.12.4 ± 0.12.3 ± 1.11.6 ± 0.510.4 ± 2.93.9 ± 0.210.2 ± 1.811.3 ± 4.1Ashes %0.12 ± 0.030.70 ± 0.50.17 ± 0.0311.7 ± 1.13.9 ± 0.40.20 ± 0.034.24 ± 0.47.3 ± 1.110.2 ± 0.2


### 2.2. Amino Acid Composition

The amino acid composition of Tilapia skin, expressed as the number of residues per 1000 total amino acid residues, is reported in Table 2. Fish skin is mainly composed of type I collagen (60–70% of dry weight) [35]. In accordance with this, Tilapia skin amino acid composition is similar to skin-derived type I collagen [8,36]. According to literature data on Tilapia skin and other fish species amino acid composition [28,36], both ST and BT were found to be rich in glycine, proline, and hydroxyproline. Thus, as well known [1,37,38], glycine was the most abundant amino acid, followed by alanine, proline, threonine, glutamate, aspartate, and hydroxyproline. No statistically significant differences were found for alanine, glycine, methionine, serine, phenylalanine, aspartate, glutamate, lysine, histidine, tyrosine, and arginine. Contrarily, valine, leucine, isoleucine, and threonine were found to decrease with age, while proline and hydroxyproline were found to increase with age. Fish skin is composed mainly of type I collagen and to a minor extent of types III and V collagen [39,40]. Since the primary structure of type I collagen should not undergo changes with age, the variation of the content of these amino acids could be ascribed to the alteration of the amount of other collagen types present in fish skin.

The imino acid content is correlated with collagen helix rigidity and consequently to the protein thermal stability and mechanical properties [41]. Unfortunately, the correlation between the imino acid content and the thermal resistance has not always been confirmed [36,42]. Moreover, contrasting results exist about the influence of fish living conditions and age on the imino acid content. Rosmawati et al. [43] reported the age did not impact on skin hydroxyproline content [44]. Love et al. reported how starvation was found to be responsible for an increased hydroxyproline content in cod skin [42]. Thus, contrarily to Rosmawati et al., our results revealed an imino acid content in line with the literature [28] and strongly affected by age, since an increase in its content from 145 ± 26 to 220 ± 15 residues was registered. The increase in the hydroxyproline content could be ascribed to the increase in the protein rigidity, and, thus, of its thermal stability and mechanical properties with age. However, the increment of skin mechanical properties could be ascribed not to the increase in the hydroxyproline content but to the increase in collagen molecules crosslinking degree that is known to increase with age [45]. However, this phenomenon is reduced in marine species, since crosslinked tissues are reported to be annually renewed and, thus, highly crosslinked proteins are not commonly found in fish [46,47]. Conversely, a more recent study by Muyonga et al. elucidated the presence of a higher molecular order content in older fish, probably because of a higher protein–protein linkages content [48]. A study reported that feeding, more than age, could affect collagen crosslinking, since starved fish were found to have more crosslinked collagen than well-fed fish [42].

### 2.3. Fatty Acid Composition

Tilapia skin fatty acid composition was quantified by GC-MS. A total of 34 fatty acids were mainly identified and are reported in Table 3. Monoenes (65% ST, 53% BT) were the most represented, followed by saturated fatty acids (29% ST, 32% BT) and polyenes (6% ST, 15% BT). Compared to the literature, a very high content of saturated fatty acid and a very low content of polyenes was found [28]. Among them, the most represented were palmitic (C16:0) and oleic acid (C18:1n9c), followed by palmitoleic acid (C16:1), octadecanoic (C18:0), (C18:1t), myristic acid (C14:0), linoleic acid (C18:2n6c), and cetoleic acid (C20:1n9) for both ST and BT [28,49]. In particular, C18:1n9c was higher than the literature, while C18:2n6c and C18:0 were lower [28]. The content of the others’ fatty acid was registered to be below 1%. Among them, C14:1, C18:3n6, C18:3n3, C20:2n6, C20:3n6, C20:3n3, C20:5n3, C21:1n9, and C22:0 were found to be lower than the literature, while C15:0 and C20:1n9 were higher [28]. Age was found to influence fatty acid content. In particular, statistically different values were reported for C14:0, C15:0, C16:1, C18:1n9c, C18:2n6c, C18:3n6, C18:0, C20:3n3, and C23:0. It seems that with age C14:0, C15:0, C16:1, C18:2n6c, C18:3n6, C18:0, C20:3n3, and C23:0 synthesis was upregulated, while C18:1n9c expression was downregulated.

### 2.4. Thermal Properties

DSC is a powerful technique for the determination of biopolymers’ hydrothermal stability [41,50]. Thermograms of ST and BT skin are reported in Figure 1. Taking into account that type I collagen is the main constituent of fish skin [24,25,26,27,28,31], the single endothermic phenomenon detected within the temperature range of 0–100 °C could be ascribed to the denaturation temperature (T_d_) of type I collagen. The enthalpy variation while heating is due to the progressive break of type I collagen inter-chain hydrogen bonds of its right-handed superhelix that brings to the triple helix unfolding in random chains (endothermic reaction) [41,51]. The determined endothermic peaks (T_d_) were found to be within the temperature range of 0–100 °C, at approximately 64.5 ± 0.2 °C for ST and 66.2 ± 0.4 °C for BT (*p* < 0.01), similar to the result reported by us for Tilapia skin [51]. Slight statistically significative differences were also found for the enthalpy required for the transition that was found to be 16.5 ± 1.3 J/g for ST and 19.8 ± 1.5 J/g for BT (*p* < 0.05). Thus, both T_d_ and ΔH were found to be influenced by age, since an increment of both was registered. The higher T_d_ and ΔH required to denature skin proteins (i.e., collagen) could be ascribed to a higher number of protein–protein linkages [48]. Unfortunately, to the best of our knowledge, few data were found on the DSC analysis on animal skin, and no data were found on Tilapia skin or other fish skin in addition to ours. 

### 2.5. Three-Dimensional Structural Investigation

Histological analysis is a destructive test usually performed on tissues sections that were previously formalin-fixed, paraffin-embedded, and then cut. In order to avoid tissues processing and sectioning, mCT offers a valid non-destructive and faster alternative for the 3D analysis of tissues samples while preserving their natural 3D structure. The coronal (X-Z), sagittal (Z-Y), and transversal (X-Y) reconstructions of ST and BT skin samples are reported in Figure 2. Herein can be appreciated Tilapia skin structure composed of an outer layer of epidermidis and an inner layer of dermis [52]. It is evident that BT (317 ± 60 µm) is significantly thicker than ST (207 ± 20 µm) (*p* < 0.001), as well as appearing to be characterized by greater porosity and greater spatial organization, with horizontally and longitudinally distributed collagen bundles within the dermis [40,52,53]. The thickness of the epidermis varies greatly depending on the part of the body, age, sex, stage of reproductive cycle, and environmental stresses [54]. In this case, it is clear that dermis thickness increases with age.

Moreover, 3D rendering reconstruction of mCT scans of ST and BT in colored scale according to density (Figure 3) allowed to better observe Tilapia skin structure. In particular, epidermis looked like a thin, flat, and denser layer, while dermis seemed to be a less dense structure [54].

### 2.6. Two-Dimensional Structural Analysis

Tilapia skin structural organization was investigated by observing its sagittal plane in a dehydrated and hydrated state by means of SEM. ST and BT imaging in the dehydrated state did not give any additional information than did mCT analysis, aside from confirming skin multilayer structure and the significative thickness difference (Figure 4a,b). Conversely, SEM imaging of samples in the hydrated state allowed to resolve the layout of the wavy-like collagen fascicles (Figure 4c–f). Both ST and BT were characterized by a thin epidermal layer and a wide dermal layer made of a connective tissue where collagen fibers were well organized in compactly arranged bundles and distributed in a parallel pattern [35,52]. A different fibril diameter and organization was observed according to age. ST was found to be characterized by a homogeneous fiber distribution through the entire skin dermis thickness that was found to be approximately 5.8 ± 0.7 μm. BT was characterized by a peculiar fiber distribution that was found to be approximately 8.8 ± 1.8 μm in most parts of the skin dermis with a progressive reduction to approximately 2.3 ± 0.9 μm in the skin dermis layers nearest to skin epidermal layer [40,52]. The collagen fiber bundles thinning and shortening in the subepidermal layer could be responsible for the stiffening of the fish skin’s external parts. This change in collagen fiber organization with age could be ascribed to the fish skin’s functional behavior of impact resistance to external agents that becomes even more evident with age.

### 2.7. Ultrastructural Analysis

In a typical 2D diffraction pattern of type I collagen (Figure 5a), the diffraction signals are located along two main directions: the equator (Figure 5a, black arrow), related to the lateral packing of collagen molecules, and the meridian (Figure 5a, red arrow), corresponding to the distances between two adjacent amino acidic residues along the molecular axis of the triple helix, the helical rise, and its pitch [55,56].

As shown in Figure 5b, 2D WAXS collected on ST and BT displayed a particular “four-lobes”-shaped intensity distribution, the marker of a spatial distribution characterized by a preferred collagen molecule orientation along two specific directions, marked with black arrows (Figure 5b), within the tissue. The intensity of the “four-lobes”-shaped signal is greater in the q-range corresponding to the equatorial diffraction signal, q = 0.5 ± 0.2 Å^−1^, in both BT (Figure 5d, black profile) and ST (Figure 5d, green profile). Thus, selecting the aforementioned q-range, the intensity distribution along the azimuth of the equatorial peak for both samples was obtained (Figure 5d). Four peaks, related to the four lobes in the diffraction pattern, were identified, and, to inspect the order of molecules within the tissue, analyses of the Full-Width-at-Half-Maximum (FWHM) of all of them were performed. Data showed that the degree of orientation of BT (Figure 5d, black profile) is slightly higher than in ST. Thus, triple helices are organized in a more ordered way in BT than in ST, probably ascribable to the increase in crosslinks with the age that tighten collagen structure. On the other hand, it appears that the characteristic “four-lobes” signal observed in fish skin is independent of age. It could be related to organization of the collagen bundles inside the thickness of the dermis, particularly to the orthogonally oriented distribution of collagen within the dermis layers, that appeared to have a slightly higher organization in BT than in ST.

Figure 6 shows the radial integration along the meridional and equatorial directions (Figure 6b,c). No differences in the meridional peak position, q = 2.15 ± 0.03 Å^−1^ corresponding to d = 2.9 ± 0.04 Å, or in the equatorial peak position, q = 0.52 ± 0.03 Å^−1^ corresponding to d = 12.08 ± 0.07Å, were identified, showing that the axial periodicity of triple helices and the distance between helices in their lateral packing within the tissues do not depend on the age of the fishes.

SAXS measurements were performed for evaluation the supramolecular arrangement of type I collagen, in particular, the presence of the nanofibrillar structures, the axial periodicity of the electron density distribution along the nanofibrils, and their orientation in the tissues. From the 2D SAXS patterns, the periodic repetition of the diffraction signals is observed in both BT (Figure 7a) and ST (Figure 7b). Folding the patterns in 1D diffraction profiles (Figure 7c,d), the peaks related to the electron density periodic distribution are visible and marked with black dotted lines. A slight intensity increase in the electron density distribution in ST was observed (Figure 7c) compared to BT (Figure 7d). This could be due to the increase in fibrils randomly oriented and layers in dermis with time, process also known as “coarse collagen” deposition [51].

### 2.8. Mechanical Properties

The mechanical properties of Tilapia skin were evaluated in a hydrated state in two mutual orthogonal directions, i.e., parallel (//) to fish backbone (longitudinal, i.e., parallel to Z axis) and perpendicular to it (⊥ or transverse, i.e., parallel to X axis). The constitutive bond of ST and BT was different if the direction of the load was longitudinal or transverse. As shown in Figure 8a,b, in the former case, for both fish sizes, an elastic behavior up to the maximum stress followed by a non-elastic region until break can be observed. Table 4 summarizes all investigated mechanical parameters.

The shape of transverse stress–strain curves (Figure 8c,d) suggested that a change in the microstructure occurs, depending on the contribution of collagen fibers and on the strain capability of the isotropic matrix tissue. The stiffness below a strain of 2–10% for ST and 2–5% for BT is very low, while the stress–strain curves showed an upper curvature associated with re-orientation of stiffer collagen fibers in the load direction. The modulus in the stiffening portion of the curves increases from 1.4 ± 0.4 MPa to 27.5 ± 7.0 MPa for ST and from 3.7 ± 1.7 MPa to 18.6 ± 6.9 for BT. The peak stress of the skin of BT and ST in the transverse direction is comparable to the one in the parallel direction but lower strains to break were achieved. This behavior can depend on the microstructure developed under the stress field generated during swimming, which generates tension and compressive strains while skin grew and its structure was developed.

The stress–strain behavior in parallel direction is associated to a more uniform stiffness, a higher toughness, and higher strain to break. Stress–strain curves of BT and ST showed a similar shape for each direction (parallel or transverse), even if differences in maximum stress and strain to break must be noted. In particular, different values of strain to break were measured for BT and ST, with a more brittle behavior of BT, probably related to the development with age of less deformable tissues. Moreover, while T value was found to not be affected by age in the ⊥ direction (*p* > 0.05), it was found to be inversely proportional to age since a significative reduction in its value was registered (*p* = 0.003).

A similar mechanical behavior was observed also for Chinese sturgeon and striped bass skin, where different stress–strain curves were obtained into the two orthogonal directions [53,57]. However, compared to Tilapia skin, their mechanical properties were revealed to be significantly different [53,57].

## 3. Discussion

Type I collagen is the most abundant vertebrate structural protein and is the most widely used biomaterial for healthcare applications [21,36,58,59]. In the last two decades, fisheries and fish industries by-products have started to be recovered for the extraction of type I collagen because of issues related to the extraction of traditional mammalian tissues [8]. In particular, special attention has been paid to by-products from fish bred in aquaponic plants. The valorization of aquaponic fish wastes as sources of biopolymers would make the derived materials not only eco-friendlier but also particularly attractive in terms of profitability and cost effectiveness. Moreover, the possibility to tune growth conditions and produce hazard-free commercial products with controlled and reproducible final properties gives them a high added value. Among fish species, Nile Tilapia is the second-most farmed species in the world and, for this reason, was selected for study. In particular, skin, the waste tissue with a higher collagen content, is commonly chosen as extraction source. However, to the best of our knowledge, no studies have been carried out to investigate, in depth, the structural and compositional age-related differences in fish skin with the final aim of selecting the most advantageous fish size for collagen extraction. 

In this work, the impact of age on the chemical, physical, and structural properties of Tilapia skin was evaluated with the aim of selecting the condition that best lends to the extraction of type I collagen for biomedical applications, based on the known fact that the properties of the original tissue have a significant impact on those of the final product. The investigation of skin properties according to age was also investigated with the aim of finding age-related differences that could allow to tune extracted collagen properties. 

First of all, skin structure was observed by mCT, confirming the presence of the thin, flat, and denser epidermis layer and the porous-like, less dense dermis layer, according to the literature on Tilapia skin [40,52,54]. The dermis thickness difference was evident, since BT was found to be thicker than ST. SEM imaging allowed to deeply investigate the morphology of Tilapia skin dermis, characterized by compactly arranged and parallelly distributed wavy-like collagen fiber bundles [35,52]. A different fibril diameter and organization was observed according to age. While ST was found to be characterized by a homogeneous fiber distribution through the entire skin dermis thickness (5.8 ± 0.7 μm), BT was characterized by a peculiar fiber distribution that was found to be higher and comparable in most parts of the dermis (8.8 ± 1.8 μm), except in the subepidermal layer, where they were found to be thinner and shorter (2.3 ± 0.9 μm) [40,52].

Fish skin general composition results were found to be in accordance with recent studies on Tilapia skin (50–70% water, 20–40% proteins, 2–15% fat, 0.1–12.0% ashes) [28,29,30] and to be strongly influenced by age. In particular, water and ash contents were found to be directly proportional to age, with a significative increase of approximately 13% for the water content and of approximately 0.6% for the ashes content. Conversely, the protein and lipid contents were found to decrease approximately 8% and 10%, respectively, according to age. Although this result was contrary to what was reported by Muyonga et al. [32], it was in accordance with mCT analysis, in which a higher porosity of the dermis was clearly visible. 

Tilapia skin fatty acid investigation identified a total of 34 fatty acids. Monoenes (65% ST, 53% BT) were the most represented, followed by saturated fatty acids (29% ST, 32% BT) and polyenes (6% ST, 15% BT). Compared to the literature, a very high content of saturated fatty acid and a very low content of polyenes was found [28]. Among them, the most represented were palmitic and oleic acid, followed by palmitoleic, octadecanoic, myristic, linoleic, and cetoleic acid for both ST and BT [28,49]. Age was found to influence fatty acid content. In particular, C14:0, C15:0, C16:1, C18:2n6c, C18:3n6, C18:0, C20:3n3, and C23:0 synthesis was upregulated with age, while C18:1n9c expression was downregulated.

A more in-depth investigation on skin amino acid composition allowed to state that both ST and BT were found to have the typical Tilapia skin amino acid composition [28,36] and that it was affected by age. Indeed, valine, leucine, isoleucine, and threonine were found to decrease with age, while proline and hydroxyproline were found to increase with age. In particular, the increase in the hydroxyproline content could be ascribed to an increase in collagen molecule rigidity which translates into an increase in their thermal and mechanical properties [45]. This hypothesis was also supported by DSC analysis results, where a slight but significative increase in both T_d_ and ΔH was found according to age. However, T_d_ and ΔH increase could be ascribed not to the increase in the molecular unit rigidity but to an increase in collagen intermolecular crosslinking degree, that is known to increase with age.

The analysis of mechanical properties of Tilapia skin confirmed the evidence which has emerged from previously discussed analyses. The most relevant finding is given by the completely different shape of stress–strain curves in the two directions that is due to the physiological mechanical stress to which fish skin is subjected during swimming. Indeed, because of their peculiar tail-bending locomotion pattern, fish skin acts as an ‘external tendon’ that transmit the force generated by longitudinal muscles to the tail. Accordingly, the parallel direction would be more subjected to stress than the perpendicular and, thus, have a higher toughness and maximum stress at higher strain values. Instead, skin deformation in the perpendicular direction can determine a re-orientation of collagen fibers as suggested by the shape of stress–strain curves, characterized by an upper curvature. Additionally, age was found to influence Tilapia skin mechanical behavior. Indeed, ST skin compared to BT skin showed higher properties in the parallel direction (toughness, strength, strain to break). The properties obtained testing BT and ST skins in the transverse direction are closer. In support of the results deriving from the mechanical tests, SEM analysis demonstrated a strictly functional structural variation linked to age. Collagen fiber bundles thinning and shortening in the subepidermal layer with age could be responsible for the stiffening of the fish skin’s external parts necessary to better withstand external agents.

WAXS and SAXS measurements demonstrated that there was no effect of age at the molecular scale. The helical pitch was preserved. However, triple helices are organized in a more ordered way in BT than in ST, probably ascribable to the increase in crosslinks with age that tighten collagen structure, as also confirmed by all other analyses. This could result in a more brittle mechanical behavior of the skin of BT, as observed above. 

## 4. Materials and Methods

### 4.1. Materials

Fish skins were peeled off Tilapia specimens bred in the pilot aquaponics plant of the “Urban Farming Lab” of the Dept. of Innovation Engineering (University of Salento, Lecce, Italy), where they were distributed in glass aquaria filled with dechlorinated water and supported with two oxygen pumps. Skins were harvested when the small (length: 25.0 ± 1.1 cm, height: 9.4 ± 0.9 cm, weight: 320.8 ± 35.8 g, ST) and the big (length: 31.6 ± 0.6 cm, height: 10.9 ± 0.9 cm, weight: 606.7 ± 19.7 g, BT) adult size were reached. Non-fish-skin components (i.e., scales, fillet) were immediately eliminated with a knife. Then, fish skins were rinsed three times with distilled water and stored at −20 °C in polyethylene bag. Prior to use, samples were thawed at 4 °C. 

Distilled water was obtained from Millipore Milli–U10 water purification facility from Merck KGaA (Darmstadt, Germany). N, N–dimethylformamide was provided by VWR International PBI S.r.l. (Milan, Italy). Acetic acid, chloroform, methanol, and CaCl_2_, norleucine, and N–tert–butyldimethylsilyl–N–methyltrifluoroacetamide (MTBSTFA) were purchased from Sigma–Aldrich (Milan, Italy). If not otherwise stated, all other chemicals used were of analytical grade and purchased from Sigma–Aldrich (Milan, Italy).

### 4.2. General Composition

Skin proximate composition was determined using the AOAC (Association of Official Analytical Chemists) procedure (AOAC International 2016). Moisture content was determined gravimetrically. Briefly, approximately 2 g of skin samples were heated at 105 °C until they reached constant weight [31]. Total protein content was assessed measuring nitrogen content by Dumas combustion method using PYRO Cube Elemental Analyser (EA) (Elementar, Hanau, Germany). In particular, 0.6–0.8 mg of dry skin was accurately weighed and sealed in tin capsule by microbalance, Sartorius CP2 P–F (Sartorius AG, Goettingen, Germany). After complete combustion, reduction, purification, and detection, the nitrogen content of fish skin samples was obtained through the Pyrocube Software V 4.0.9. Crude protein content was derived by multiplying nitrogen content by 6.25 [60]. Crude lipid content was determined by Folch method [61,62]. Briefly, 0.250 g of dried and ground samples were homogenized by mean of an IKA T25 Digital ultra Turrax (IKA^®^-Werke GmbH and Co. KG, Staufen im Breisgau, Germany) with a 2:1 (*v*/*v*) chloroform–methanol mixture at 15,000 rpm for 3 min. Subsequently, the homogenate was filtered through Whatman filter paper (CAT No. 1441-070) and mixed with 0.2 of its volume of 0.05% (*w*/*v*) CaCl_2_. The mixture was separated into two phases by centrifugation at 2400 rpm for 20 min and the upper phase was discarded. The lower phase was washed 3 times with a small amount of pure upper phase solvent consisting of a 3:48:47 (*v*/*v*) mixture of chloroform–methanol–saline solution (0.02% (*w*/*v*) CaCl_2_). Finally, the resulting lower phase was dried for subsequent gravimetric quantification. Ash content was determined gravimetrically by the incineration method [31,43,60]. Briefly, 2.0 g sample was burned in a muffle furnace at 550 °C for six hours, in a previously tared porcelain capsule. Then, ashes were cooled down in a desiccator and weighed.

### 4.3. Amino Acid Composition

The amino acid composition of fish skin was investigated by means of GC–MS as described elsewhere [36,63,64]. After hydrolysis in 6 N hydrochloric acid for 24 h at 120 °C, samples were 0.2 μm filtered and freeze-dried. Norleucine (5 μL of a 100 ng/μL norleucine solution) was added as an internal standard. Then, the dry residues were reconstituted with 70 μL of N, N-dimethylformamide, and 20 μL of MTBSTFA. Derivatization with MTBSTFA was performed at 100 °C for 60 min. After cooling the solution at room temperature for 5 min, 1 μL of the solutions was injected in spit mode (split ratio 10:1). Samples were run on a GC–QqQ–MS (Bruker 456 gas chromatograph coupled to a triple quadrupole mass spectrometer Bruker Scion TQ) equipped with an autosampler (GC PAL, CTC Analytics AG). The GC was operated at a constant flow of 1.0 mL/min, and analytes were separated on an HP 5 MS capillary column (50 m with a 2 m guard column, inner diameter 250 μm, and film thickness 0.25 μm) [36,63,64]. After injection, the oven was kept at 60 °C for 1 min. Then, a temperature of 320 °C was reached through a gradient of 10 °C/min and held for 10 min. The total run time was 37 min. The mass detector was operated at 70 eV in the electron impact ionization mode scanning the mass range 50–550 Da. The ion source and transfer line temperatures were 230 °C and 280 °C, respectively. Bruker MS Workstation 8.2 software was used to acquire chromatograms, process the data, and quantify amino acids. Samples were run thrice.

### 4.4. Fatty Acid Composition

The lipid composition of Tilapia skin was investigated by mean of GC–MS. Dried fish skins (5 mg) were placed in a 2 mL glass vial. An internal standard solution of 1000 mg/L n–tetratriacontane in cyclohexane (10 μL) and 1 mL of BF3 solution in methanol (10%) were added. Methylation reaction was conducted at 70 °C for 30 min and then 500 μL was added in glass vial. The methyl esters were solvent-extracted (hexane; 3 × 1 mL) by alternating vortex agitation. The extracts were collected, evaporated to dryness, and dissolved in 400 µL of hexane. The samples were analyzed on a Bruker SCION–456GC gas chromatograph equipped with PAL autosampler and coupled to a Bruker SCION TQ mass spectrometer. The sample (1 µL) was injected at 280 °C in splitless mode and methyl esters separated on a J&W DB–5ms (Agilent, Santa Clara, California, United States of America) capillary column (60 m, 0.25 mm i.d., and 0.25 μm film thickness) using helium (99.9995%) as carrier gas at constant flow (1.2 mL/min). The oven temperature program was as follows: 50 °C (1 min), up to 320 °C at a rate of 6 °C/min (10 min). The MS transfer line temperature and the MS ion source temperature were 280 °C and 230 °C, respectively. The EI spectra were acquired in full scan mode in the range m/z 50–550. FAME quantification was performed using FAME 37 and FAME C4–C24 (both from Supelco) CRMs as external standards. Samples were run thrice.

### 4.5. Thermal Properties

A Q2000 Series DSC from TA Instruments (New Castle, DE, USA) was employed for DSC measurements. Dry skin samples were accurately weighed (5–7 mg) into aluminum pans, hermetically sealed, and scanned from 5°C to 100°C at 5°C/min in inert nitrogen atmosphere (50 mL min^−1^) [65,66]. An empty aluminum pan was used as a reference. The temperature at which occurred the endothermic phenomenon was measured as the mid-point of the corresponding endothermic peak and the area under the peak was calculated to estimate the enthalpy of the transition [41,66]. The obtained data were analyzed using OriginPro software (Origin–Lab version 8, Northampton, MA, USA). Three fish skin pieces belonging to three different fish specimens were tested for each sample type. 

### 4.6. Three-Dimensional Structural Investigation

Tilapia skin 3D organization was visualized by mean of Skyscan 1072 X-ray microtomography system (Bruker, Munich, Germany) on the central area on freeze-dried skin samples of 8 mm diameter. The operational parameters of 26 kV and 151 μA for the sealed X-ray tube were used to carry out two-dimensional image analysis with the resolution of 2.98 μm. A radiographic projection of the sample was repeatedly taken for each angle during the scanning while the sample was turned with a step rotation of 0.3° around the Z-axis. The scanning rotation was of 180°. No filter was applied. X-ray absorption radiographs were acquired through the specimen and a slice reconstruction was performed by Bruker NRecon software (Version 1.6.9.8). Finally, the datasets were processed by Bruker CTan (version 1.20.8.0) software, for a fine alignment with the reference axis system, and by Bruker DataViewer (version 1.6.0.0) and CTVox (version 3.3.1) software, for 3D analysis of sample microstructure based on the absorption coefficients of X-rays.

### 4.7. Two-Dimensional Structural Analysis

Tilapia skin structural organization was investigated by observing its sagittal plane in dehydrated and hydrated state with a SEM EVO^®^ 40 (Carl Zeiss AG, Oberkochen, Germany). Thus, in order to investigate skin structural organization in dehydrated state, ST and BT samples were freeze-dried, mounted on an aluminum stub, and directly observed at SEM with an accelerating voltage of 20 kV. In order to investigate skin structural organization in hydrated state, −20 °C frozen skin samples of 0.5 × 1.0 cm were thawed at 4 °C and formalin 10% (pH 7.5) fixed for 2 h at room temperature. After two washing cycles of 10 min with tap water, samples were dehydrated in graded alcohol series (50%, 70%, 90%, 96%, 100% ethanol, 15 min, respectively) and cleared with xylene for 10 min. Then, samples were paraffin-embedded and sectioned at 5 μm thickness with a PowerTome XL Ultramicrotome (RMC Products by Boeckeler Instruments Inc., Tucson, AZ, USA). Sections representative of Tilapia skin thickness were cleared with xylene for 10 min to remove paraffin, double-washed with ethanol 100% for 5 min, air-dried, mounted on an aluminum stub, and directly observed at SEM with an accelerating voltage of 20 kV.

### 4.8. Ultrastructural Analysis

WAXS and SAXS analyses were performed at the X-ray Micro Imaging laboratory (XMI-LAB) IC-CNR, Bari [67] on Tilapia skin samples. The laboratory is equipped with a Fr-E + SuperBright rotating copper anode micro-source (λ = 0.154 nm, 2475 W), which is coupled by a multilayer focus optics (Confocal Max-Flux CMF 15-105) to a SAXS/WAXS three-pinhole camera. WAXS data were collected using an image detector (IP) (250 × 160 mm^2^, with 100 µm effective pixels) placed at around 10 cm distance from the sample, giving access to a range of scattering vector moduli (q = 4 πsinθ/λ) from 0.3 to 3.5 Å^−1^, corresponding to a 1.8–21 Å spacing range in the direct space. Data were digitalized by an off-line RAXIA reader and elaborated by SAXGUI and SUNBIM software [68]. SAXS measurements were collected by using a Triton 20 gas-filled proportional counter (1024 × 1024 array, 195 μm pixel size) placed at around 2200 mm distance from the sample, giving access to a range of scattering vector moduli from 0.006 to 0.2 Å^−1^, corresponding to a 3–105 nm range in the direct space. Both SAXS and WAXS data were acquired on three different points per sample condition, each point for 3600 s. Slices of ST and BT (1 × 1 cm^2^) were kept in Ultralene^®^ sachets together with a drop of distilled water for X-ray acquisition. The sachets were then sealed in order to preserve the hydration state of the tissues. Ultralene^®^ is a thin film that confers the uniform transmission of X-rays. Because of its chemical and heat strength and good X-ray transmission, it is commonly used in X-ray analysis with liquid and wet samples. The sealed sachets of Ultralene^®^ with the fish skins were mounted on a sample holder and placed in a chamber in vacuum (0.1/1 mbar) during acquisition.

### 4.9. Mechanical Properties

The constitutive bond of Tilapia skin was evaluated in hydrated state using a ZwickiLine universal testing machine (Zwick/Roell, Ulm, Germany) equipped with a loading cell of 1 kN. Wet skin samples were 5 × 20 mm cut following two orthogonal directions: the parallel group (//) was the one in which skin samples were cut along the fish horizontal long axis, while the perpendicular group (⊥) was the one in which skin samples were cut orthogonally to the fish horizontal long axis. Then, samples were equilibrated in 0.01 M PBS at room temperature for 1 h, clamped, and tensile-tested under displacement control until failure with a preload of 0.1 N and a load speed of 0.2 mm/s [66]. The Young modulus (E), the stress at break (*σ*_max_), the strain at break (*ε*_r_), and the toughness (T) were measured. E was calculated as the slope of the linear elastic region of the stress–strain curve at low strain values (in the range 1–5%). T was calculated as the area under the stress–strain curve. The thickness and width of wet specimens was measured using a Dino-Lite digital microscope (AnMo Electronics Corporation, New Taipei City, Taiwan). Six samples were analyzed for each sample type. 

### 4.10. Statistical Analysis

All data were expressed as mean ± the standard deviation. Statistical significance of experimental data was determined using Student’s *t*-test. Differences were considered significant at *p* < 0.05.

## Figures and Tables

**Figure 1 ijms-24-01938-f001:**
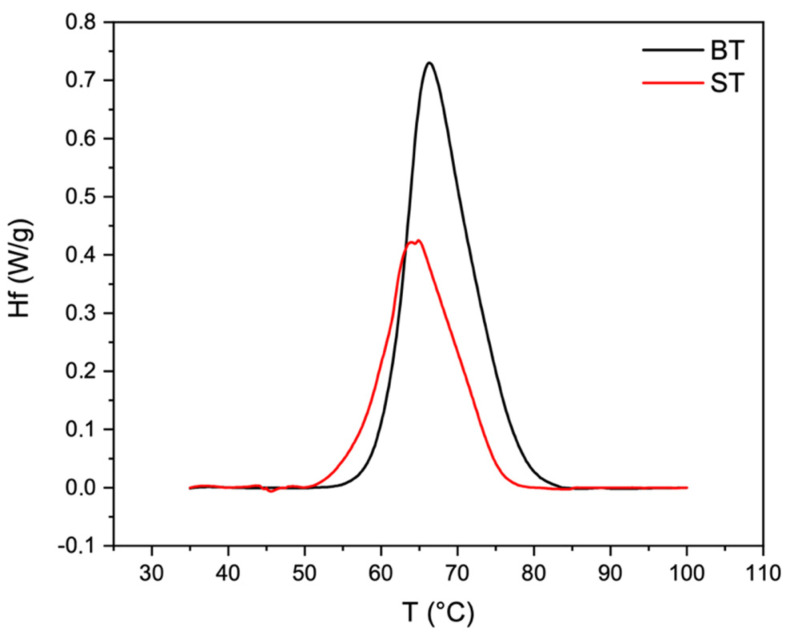
Representative DSC thermograms of ST and BT.

**Figure 2 ijms-24-01938-f002:**
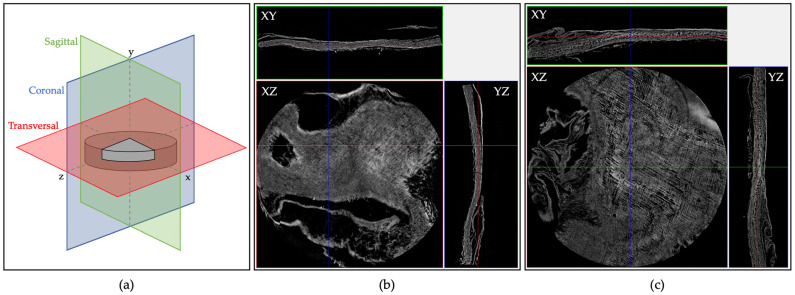
Coronal (Z-Y), sagittal (X-Y), and transversal (X-Z) view (**a**) of ST (**b**) and BT (**c**), where Z was the axis parallel to the fish backbone. Tilapia skin dermis can be observed in the transversal reconstruction. Sample size: 8 mm.

**Figure 3 ijms-24-01938-f003:**
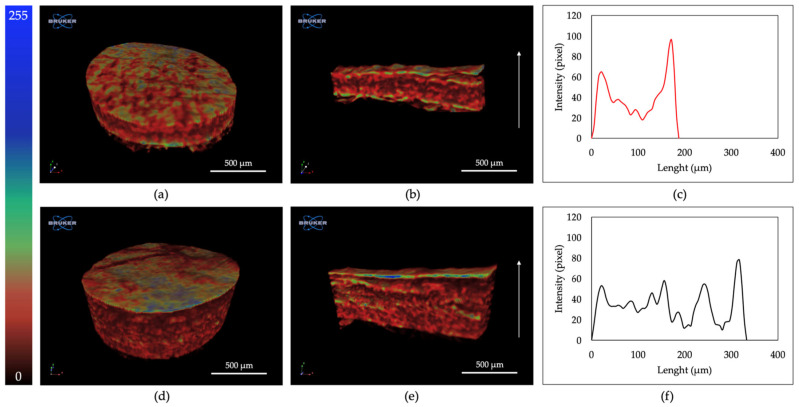
Three-dimensional rendering reconstruction of mCT scans of ST (**a**) and BT (**d**), their coronal section (**b**,**e**, respectively) in colored scale, and their relative intensity profile (**c**,**f**, respectively).

**Figure 4 ijms-24-01938-f004:**
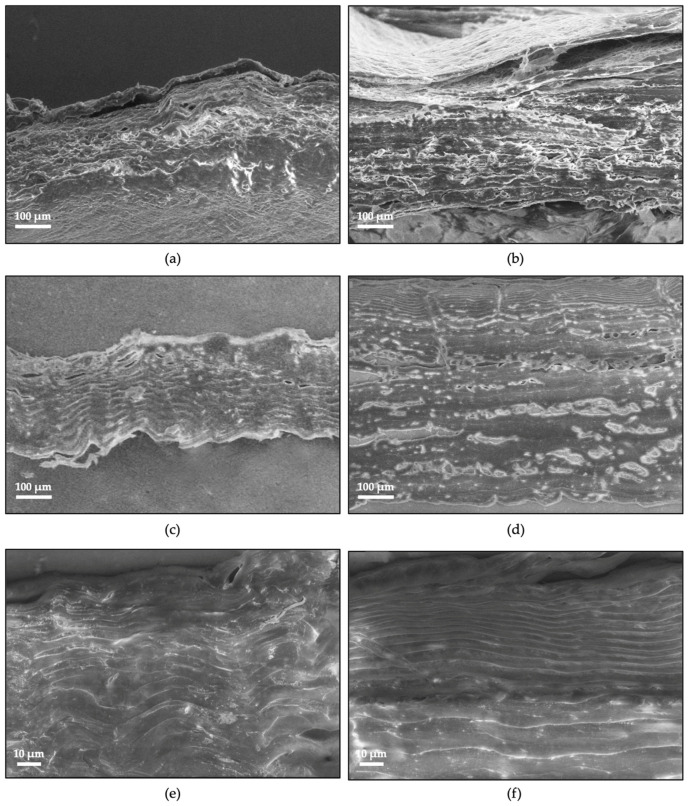
SEM imaging of ST (**a**) and BT (**b**) in dehydrated state, representative of the different skin thickness according to age (Magnification: 300×) and of ST (**c**,**e**) and BT (**d**,**f**) in hydrated state, showing the wavy-like structure of Tilapia skin collagen fibers (Magnification: 500× and 2000×).

**Figure 5 ijms-24-01938-f005:**
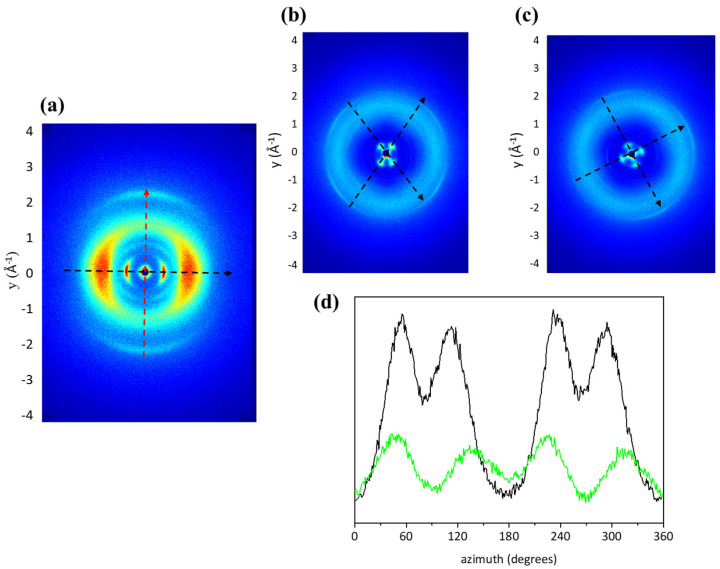
(**a**) A typical 2D WAXS diffraction pattern on type I collagen from animal tissue, in particular equine tendon, in which the equatorial (black arrow) and meridional (red arrow) directions are clearly distinguished. In the 2D diffraction patterns of BT (**b**) and ST (**c**), the diffraction signal is characterized by four lobes along two directions (black arrows). (**d**) The azimuthal integration along the equatorial diffraction signal of both BT (black profile) and ST (green profile) are shown.

**Figure 6 ijms-24-01938-f006:**
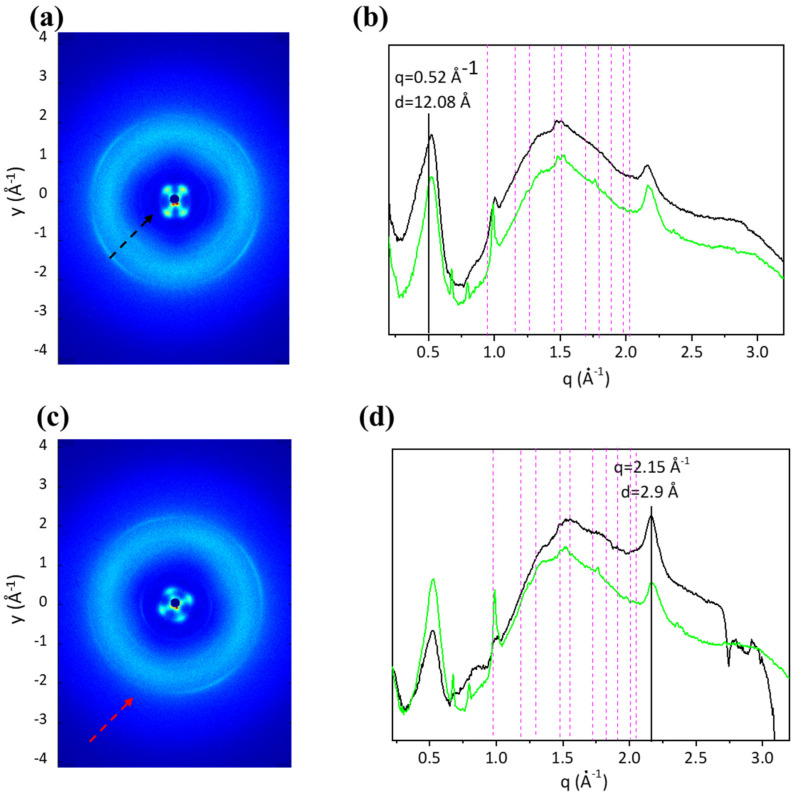
In the 2D diffraction patterns of BT (**a**) and ST (**c**) the equatorial and meridional signals are shown by black and red arrows, respectively. The radial integration along the equatorial (**b**) and meridional (**d**) direction for both ST (green profile) and BT (black profile) are shown. Magenta vertical dotted bars refer to Ultralene^®^ sachet diffraction.

**Figure 7 ijms-24-01938-f007:**
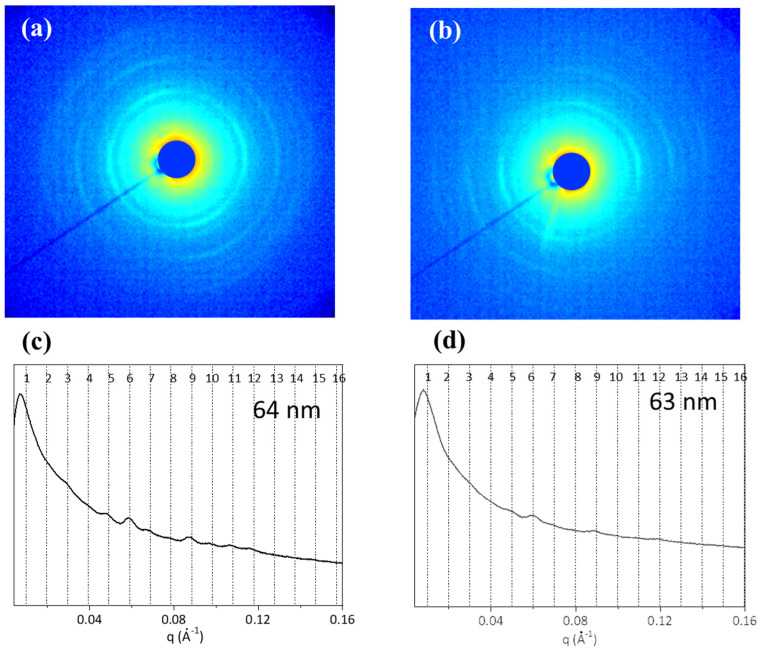
The 2D SAXS patterns of BT (**a**) and ST (**b**) are shown. In the corresponding profiles (**c**,**d**), the black dotted lines indicate the fibrillary periodicity.

**Figure 8 ijms-24-01938-f008:**
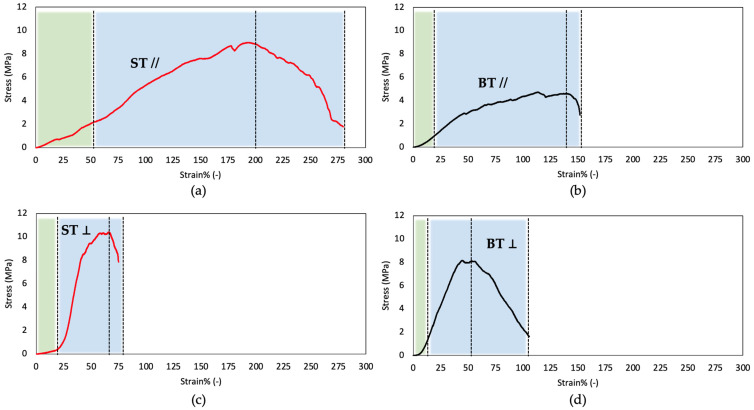
Representative stress–strain curves of ST (**a**,**c**) and BT (**b**,**d**) skin. Plots (**a**,**b**) were obtained in longitudinal direction, (**c**,**d**) in the transverse direction.

**Table 2 ijms-24-01938-t002:** Amino acid composition of ST and BT skin in comparison with literature data about Tilapia skin composition. Results are expressed as residues/1000 total amino acid residues.

Amino Acids	ST	BT	[28]
Alanine	139 ± 20	144 ± 11	157.0 ± 1.5
Glycine	287 ± 52	289 ± 15	381.1 ± 5.2
Valine	23 ± 1	20 ± 1	17.5 ± 0.8
Leucine	36 ± 8	23 ± 6	24.7 ± 0.7
Isoleucine	15 ± 2	9 ± 1	9.0 ± 0.2
Proline	90 ± 13	119 ± 7	97.8 ± 9.9
Methionine	4 ± 3	4 ± 2	7.7 ± 0.5
Serine	42 ± 3	38 ± 5	30.7 ± 0.5
Threonine	61 ± 7	29 ± 6	26.8 ± 0.8
Phenylalanine	18 ± 6	15 ± 6	12.6 ± 1.1
Aspartate	59 ± 4	51 ± 7	32.0 ± 0.5
Glutamate	87 ± 4	83 ± 9	56.6 ± 0.9
Lysine	23 ± 9	18 ± 4	21.9 ± 0.4
Histidine	1 ± 1	1 ± 0	5.0 ± 0.4
Tyrosine	5 ± 3	3 ± 0	4.3 ± 0.3
Hydroxyproline	65 ± 14	95 ± 13	53.3 ± 1.0
Arginine	35 ± 5	38 ± 4	61.4 ± 1.8
Imino acids	145 ± 26	220 ± 15	151.1 ± 8.9
Tot.	1000	1000	1000

**Table 3 ijms-24-01938-t003:** Fatty acid composition % of ST and BT skin. ND = not detected. Results are expressed as mean value ± SD.

Fatty Acids	ST	BT	[28]
C14:1	0.05 ± 0.02	0.06 ± 0.01	0.30 ± 0.04
C14:0	3.05 ± 0.54	4.03 ± 0.18	3.90 ± 0.57
C15:1	0.010 ± 0.004	0.05 ± 0.05	ND
C15:0	0.52 ± 0.02	0.90 ± 0.06	0.28 ± 0.04
C16:1	5.40 ± 0.56	7.23 ± 0.96	7.25 ± 0.94
C16:0	19.81 ± 2.17	20.78 ± 0.69	19.39 ± 2.69
C17:1	ND	ND	0.53 ± 0.09
C17:0	0.22 ± 0.01	0.30 ± 0.05	ND
C18:2t	0.08 ± 0.04	0.15 ± 0.04	ND
C18:1t	3.30 ± 0.37	3.26 ± 0.23	ND
C18:1n9c	56.36 ± 2.07	43.59 ± 2.15	23.72 ± 11.37
C18:2n6c	2.78 ± 0.98	7.40 ± 0.97	10.83 ± 1.55
C18:3n6	0.005 ± 0.001	0.03 ± 0.01	0.93 ± 0.13
C18:3n3	0.13 ± 0.14	0.43 ± 0.16	8.89 ± 1.63
C18:0	4.82 ± 0.82	6.36 ± 0.47	8.16 ± 1.48
C19:0	0.07 ± 0.01	0.11 ± 0.04	ND
C20:0	0.12 ± 0.02	0.11 ± 0.09	0.31 ± 0.07
C20:1n9	1.68 ± 0.05	1.17 ± 0.33	0.88 ± 0.11
C20:2n6	0.11 ± 0.8	0.27 ± 0.14	1.07 ± 0.19
C20:3n6	0.06 ± 0.09	0.04 ± 0.02	1.41 ± 0.24
C20:3n3	0.015 ± 0.016	0.08 ± 0.03	2.74 ± 0.50
C20:4n6	0.007 ± 0.007	0.32 ± 0.27	0.31 ± 0.02
C20:5n3	0.014 ± 0.006	0.12 ± 0.13	0.90 ± 0.11
C21:1n9	0.02 ± 0.01	0.02 ± 0.06	0.44 ± 0.09
C22:0	0.013 ± 0.014	0.12 ± 0.08	0.34 ± 0.09
C22:1n9	0.05 ± 0.05	0.08 ± 0.03	ND
C22:2n6	0.007 ± 0.006	0.02 ± 0.01	0.02 ± 0.04
C22:4n6	0.14 ± 0.22	1.24 ± 0.96	ND
C22:6n3	0.05 ± 0.08	0.47 ± 0.36	ND
C23:0	0.003 ± 0.003	0.017 ± 0.012	ND
C24:0	0.02 ± 0.02	0.06 ± 0.05	ND
C24:1n9	0.02 ± 0.01	0.16 ± 0.27	0.23 ± 0.10
C24:6n3	ND	ND	7.16 ± 1.21
Saturates	28.6	31.7	32.4
Monoenes	65.2	53.1	33.4
Polyenes	6.1	15.2	34.3

**Table 4 ijms-24-01938-t004:** Mechanical properties of ST and BT. Results are expressed as mean value ± SD.

	ST		BT	
	//	⊥	//	⊥
E (MPa)	2.6 ± 0.9 (2–10%) 5.7 ± 2.0 (60–80%)	1.4 ± 0.4 (2–10%) 10.3 ± 1.3 (20–25%) 27.5 ± 7.0 (30–40%)	2.4 ± 1.2 (2–10%) 4.1 ± 1.4 (60–80%)	3.7 ± 1.7 (2–5%) 15.5 ± 3.0 (5–10%) 18.6 ± 6.9 (30–40%)
σ_max_ (MPa)	8.5 ± 1.1	8.6 ± 1.9	5.3 ± 0.7	7.2 ± 1.0
ε_r_ (%)	207 ± 75	71 ± 17	187 ± 79	53 ± 18
T (MPa)	1687 ± 286	371 ± 147	565 ± 121	477 ± 30

## Data Availability

Not applicable.

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
