# Peer review of "Age-Related Properties of Aquaponics-Derived Tilapia Skin (Oreochromis niloticus): A Structural and Compositional Study"

_ijms, 2023, doi:10.3390/ijms24031938_

Round 1

Reviewer 1 Report

The paper essentially concerns the sourcing of collagen for biomedical applications from fisheries waste. This may lead to a sensible, sustainable and cost-effective approach without compromising the ultimate quality of the end-product.

On one hand, the paper is very well conceived and shows a great care and precision in the laboratory work. On the other hand, the techniques used are not always the most appropriate ones. The greatest shortcoming is the complete lack of an ultrastructural evaluation, a thing that I see as absolutely mandatory in such a study. 

Moreover, it shows some surprising naivety in dealing with the collagen structure and properties.

In particular:

Line 110: “both ST and BT were found to  be rich in glycine, proline, hydroxyproline. Glycine was the most abundant species...”. This is nothing new since 1/3 of all fibrillar collagens residues MUST be glycine, while proline and hydroxyproline are much more frequent than in other proteins.

Line 114: “valine, leucine, isoleucine, and threonine were found to decrease with age while proline and hydroxyproline were found to increase with age”. Since the primary structure of type I collagen does not vary with age, the variation of these aminoacids presumably reflect varying amounts of other collagen types such as, by instance, types III or V, which accompany type I in most tissues.

Line 122: “Asknes et al. [28] reported that an hydroxyproline enriched diet not influenced fish hydroxyproline content”. This is not surprising since hydroxyproline cannot be incorporated as such (i.e., there is no codon for it) but comes from post-translational modification of “normal” proline residues by at least two 4-prolyl-hydroxylases and one 3-prolyl-hydroxylase.

Line 127:  “The increase of the hydroxyproline content could be ascribed to the increase of collagen crosslinking that it is known to increase with age”. Cross-links form exclusively between lysine and hydroxylisine residues and, in just one instance, a histidine residue. To my knowledge, hydroxyproline was never been related to cross-links. Reference [29] is mis-quoted: it deals indeed with cross-links, but never suggests an implication of hydroxyproline.

Line 184: “In order to avoid tissues processing and sectioning, mCT offers a valid non-destructive and faster alternative for the 3D analysis of tissues samples”. On the other hand, because of its poor spatial resolution, mCT cannot resolve the layout of the collagen fascicles, let alone the fibril diameter distribution which can be very different among different tissues and/or ages and which is an important predictor of the functional behaviour of the tissue. Scanning electron microscopy is simpler and  faster than histological analysis and has a huge advantage in resolution with respect to both mCT and histology. In this work an electron microscopical evaluation of the tissues is, in my opinion, mandatory.

Line 342: see Line 127.

Line 346: “The most relevant finding is given by the completely different shape of stress strain curves in the two directions probably due to the physiological mechanical stress at which fish skin is subjected during swimming”. This is correct. It is important to note that fish skin is something of an exception: in terrestrial animals the skin has nothing to do with the locomotor apparatus and its collagen fibres are wavy and almost isotropically oriented. In fishes, by contrast, because of their peculiar tail-bending locomotion pattern, the skin acts as a sort of external tendon to transmit to the tail the force of longitudinal muscles: its fibrils are mostly longitudinally oriented (and this explains the different stress-strain curves) and their ultrastructure is more reminiscent of mammalian tendon than of skin.

Author Response

Response to Reviewer 1 Comments

The paper essentially concerns the sourcing of collagen for biomedical applications from fisheries waste. This may lead to a sensible, sustainable and cost-effective approach without compromising the ultimate quality of the end-product. On one hand, the paper is very well conceived and shows a great care and precision in the laboratory work. On the other hand, the techniques used are not always the most appropriate ones. The greatest shortcoming is the complete lack of an ultrastructural evaluation, a thing that I see as absolutely mandatory in such a study. Moreover, it shows some surprising naivety in dealing with the collagen structure and properties. In particular:

Point 1: Line 110: “both ST and BT were found to be rich in glycine, proline, hydroxyproline. Glycine was the most abundant species...”. This is nothing new since 1/3 of all fibrillar collagens residues MUST be glycine, while proline and hydroxyproline are much more frequent than in other proteins.

Response 1: We thank the Reviewer for her/his remark. The aim of the reported sentence was not to add something new but just to state and confirm what is already known. Accordingly, to stress this aspect, we slightly modified the sentence as: ‘Thus, as well known, glycine was the most abundant specie followed by alanine, proline, threonine, glutamate, aspartate and hydroxyproline.’

Point 2: Line 114: “valine, leucine, isoleucine, and threonine were found to decrease with age while proline and hydroxyproline were found to increase with age”. Since the primary structure of type I collagen does not vary with age, the variation of these aminoacids presumably reflect varying amounts of other collagen types such as, by instance, types III or V, which accompany type I in most tissues.

Response 2: We thank the Reviewer for her/his suggestion. Accordingly, we added an explicative sentence to explain the variation of the content of valine, leucine, isoleucine, and threonine could be attributed to the variation of the content of other collagens type present in fish skin, like III and V, that is: ‘Fish skin is composed mainly of type I collagen and in a minor extent of types III and V collagen [10.1111/j.1365-2621.2001.tb11325.x, 10.1007/s10561-017-9681-y]. Since the primary structure of type I collagen should not undergo changes with age, the variation of the content of these amino acids could be ascribed to the alteration of the amount of other collagen types present in fish skin.’

Point 3: Line 122: “Asknes et al. [28] reported that an hydroxyproline enriched diet not influenced fish hydroxyproline content”. This is not surprising since hydroxyproline cannot be incorporated as such (i.e., there is no codon for it) but comes from post-translational modification of “normal” proline residues by at least two 4-prolyl-hydroxylases and one 3-prolyl-hydroxylase.

Response 3: We thank the reviewer for her/his comment. Accordingly, we removed the sentence since it did not bring a significant and useful contribution to the discussion of the results.

Point 4: Line 127:  “The increase of the hydroxyproline content could be ascribed to the increase of collagen crosslinking that it is known to increase with age”. Cross-links form exclusively between lysine and hydroxylisine residues and, in just one instance, a histidine residue. To my knowledge, hydroxyproline was never been related to cross-links. Reference [29] is mis-quoted: it deals indeed with cross-links, but never suggests an implication of hydroxyproline.

Response 4: We thank the Reviewer for his/her observation. Accordingly, we corrected the sentence as: ‘The increase of the hydroxyproline content could be ascribed to the increase of the protein rigidity, and thus of its thermal stability and mechanical properties with age. However, the increment of skin mechanical properties could be ascribed not to the increase of the hydroxyproline content but to the increase of collagen molecules crosslinking degree that it is known to increase with age.’

Point 5: Line 184: “In order to avoid tissues processing and sectioning, mCT offers a valid non-destructive and faster alternative for the 3D analysis of tissues samples”. On the other hand, because of its poor spatial resolution, mCT cannot resolve the layout of the collagen fascicles, let alone the fibril diameter distribution which can be very different among different tissues and/or ages and which is an important predictor of the functional behaviour of the tissue. Scanning electron microscopy is simpler and faster than histological analysis and has a huge advantage in resolution with respect to both mCT and histology. In this work an electron microscopical evaluation of the tissues is, in my opinion, mandatory.

Response 5: We thank the Reviewer for his/her constructive criticism. In full agreement with his/her comment, SEM analysis was performed on Tilapia skin samples in dehydrated state and in hydrated (and fixed) state in order to investigate about collagen fascicles morphology, size and distribution. Accordingly, two new sections were added in the main text regarding SEM analysis materials and methods (Section 4.7), results (Section 2.6), discussion and a new figure (Figure 4). While SEM imaging in dehydrated state did not give any additional information, SEM imaging of samples in hydrated and fixed state allowed to resolve the layout of the wavy-like collagen fascicles and acquire very informative data. In particular, a different fibril diameter and organization was observed according to age. ST was characterized by a homogeneous fibers distribution through the entire skin dermis thickness that was found to be of about 5.8 ± 0.7 mm. Conversely, BT was characterized by a peculiar fiber distribution that was found to be of about 8.8 ± 1.8 mm in the most part of the skin dermis with a progressive reduction to about 2.3 ± 0.9 mm in the skin dermis layers nearest to skin epidermal layer [26,38]. The collagen fibers bundles thinning and shortening in the subepidermal layer could be responsible of the fish skin external part stiffening. This change in collagen fibers organization with age could be ascribed to the fish skin functional behavior of impact resistance to external agents that becomes even more evident with age.

Point 6: Line 342: see Line 127.

Response 6: We thank the reviewer for its instruction. According to Reviewer comments on Line 127, we modified Line 342 contents.

Point 7: Line 346: “The most relevant finding is given by the completely different shape of stress strain curves in the two directions probably due to the physiological mechanical stress at which fish skin is subjected during swimming”. This is correct. It is important to note that fish skin is something of an exception: in terrestrial animals the skin has nothing to do with the locomotor apparatus and its collagen fibres are wavy and almost isotropically oriented. In fishes, by contrast, because of their peculiar tail-bending locomotion pattern, the skin acts as a sort of external tendon to transmit to the tail the force of longitudinal muscles: its fibrils are mostly longitudinally oriented (and this explains the different stress-strain curves) and their ultrastructure is more reminiscent of mammalian tendon than of skin.

Response 7: We thank the Reviewer for her/his positive feedback and valuable comment. We enriched and strengthened the sentence reported at Line 346 according to her/his suggestions.

Reviewer 2 Report

This is a well-written article, and I recommend Publishing it after minor revision. 

Add quantitative data to abstract 

update references

compare result with similar studies. 

Author Response

Response to Reviewer 2 Comments

This is a well-written article, and I recommend Publishing it after minor revision. 

Point 1: Add quantitative data to abstract 

Response 1: We thank the Reviewer for her/his suggestion. Accordingly, we added quantitative data to the abstract. The following text has been added into the abstract:  “In particular, an increase of skin thickness (+110 µm) and of wavy-like collagen fiber bundles diameter (+3 µm) besides their organization variation was observed with age. Additionally, a preferred collagen molecules orientation along two specific directions was revealed, with a higher fiber orientation degree according with age. Thermal analysis registered a shift of the endothermic peak (+1.7°C) and an increase of the en-thalpy (+3.3 J/g) while mechanical properties were found to be anisotropic with an age-dependent brittle behavior. Water (+13%) and ash (+0.6%) contents were found to be directly proportional with age, as opposed to protein (-8%) and lipid (-10%) con-tents. The amino acid composition revealed a decrease of the valine, leucine, isoleu-cine, and threonine content and an increase of proline and hydroxyproline. Lastly, fat-ty acids C14:0, C15:0, C16:1, C18:2n6c, C18:3n6, C18:0, C20:3n3 and C23:0 revealed to be upregulated while C18:1n9c was downregulated with age.”

Point 2: update references

Response 2: We thank the Reviewer for her/his comment. Accordingly, references list was updated by adding the following: 10.1049/iet-nbt.2019.0297, 10.1016/j.jmbbm.2016.09.031, 10.1016/s0378-5173(01)00691-3, 10.1111/ijfs.16104, 10.1007/978-3-642-16555-9_3, 10.1007/s00441-009-0844-4.Collagens, 10.1002/ar.10162, 10.1016/S0378-5173(01)00691-3, doi.org/10.3390/ foods11192985, doi.org/10.3390/md20110664, 10.3389/fsufs.2021.611835, 10.3390/md20010061, 10.3390/md17080467, 10.1063/5.0036110, 10.1007/s13197-014-1652-8, 10.1111/j.1365-2621.2001.tb11325.x, 10.1007/s10561-017-9681-y, 10.1016/j.acthis.2021.151762, 10.1186/s12917-020-02693-w, 10.3390/md17080467, 10.1002/adma.201801651, 10.5772/intechopen.77051.

Point 3: compare result with similar studies

Response 3: We thank the Reviewer for her/his comment. For the best of our knowledge, no similar studies were available. However, Tilapia skin general composition, amino acid and lipid content, WAXS and SAXS profiles, besides thermal, morphological and mechanical properties results were compared with available and pertinent literature.

Round 2

Reviewer 1 Report

The authors reworked extensively the original manuscript, included new original data and pictures, and answered satisfactorily to all the objections of this reviewer. The paper is interesting and no longer shows any significant shortcoming.